# Impact of COVID-19 on Performance Evaluation Large Market Capitalization Stocks and Open Innovation

**Immas Nurhayati [1], Endri Endri [2,*](ID), Renea Shinta Aminda [1] and Leny Muniroh [1]**

[1] Management Department, Faculty of Economics and Business, Universitas Ibn Khaldun, Bogor 16162, Indonesia; immasnurhayati1@gmail.com (I.N.); renea_shinta@yahoo.com (R.S.A.); lenymuniroh@gmail.com (L.M.)

[2] Faculty of Economics and Business, Universitas Mercu Buana, Jakarta 11650, Indonesia

[*] Correspondence: endri@mercubuana.ac.id; Tel.: +62-8129204067

**Abstract:** This research is an event study that evaluates the performance of large market capitalization shares using a performance model that is adjusted to risks due to the COVID-19 outbreak. The study measured the performance of large market capitalization stocks which represented each tick size on the Indonesian Stock Exchange during the COVID-19 pandemic using the Sharpe Index, the Treynor Ratio, and Jensen's Alpha. The sample selection used a purposive sampling technique and 24 stocks were selected as samples in the study. We used the daily closing price of stocks, the Indonesia composite index, and average risk-free rate return (BI rate). By using Jensen's Alpha, this study found that FREN was the highest beta with a value of 1.8189, indicating that the index was an effective and well-diversified stock. FREN is low priced and the highest market capitalization stock in its tick size (third tier stocks). Jensen's Alpha is good for measuring the performance of large capitalization and low-priced stocks. There are eight stocks that always have negative values in each method of measuring stock performance, which indicates that these stocks underperformed during COVID-19.

**Keywords:** event study; COVID-19 outbreak; large market capitalization; risk adjusted performance; underperformed

## 1. Introduction

The current coronavirus (COVID-19) infection outbreak has a major impact on various aspects of life. COVID-19 is spreading globally, not only concerning health issues, but also affecting the economy and the environment in various ways. As of 24 December 2020, there are 222 countries, areas, or territories with cases, 77,228,903 people confirmed positive, and 1,718,470 confirmed deaths [1]. The COVID-19 pandemic has almost crippled the economy and has had an impact on various sectors. The COVID-19 outbreak has made the world economy go into recession, including Indonesia. Drastic changes in the socioeconomic life of the community have changed the buying and selling interactions in the market. Some industries have experienced deep downturns, and others have benefited from the disaster, but as a whole, the Indonesian economy has experienced quite a frightening contraction. Before the pandemic occurred, the global economy and the national economy were in a good condition, stable and prospective for investing. This can be seen from the IDX composite at the beginning of January at 6300 and national economic growth at the level of 5.5%. During the pandemic, the IDX composite trend declined to reach 4000. This decline was inseparable from investor sentiment, most of which were risk averse and risk neutral, which attracted funds from the capital market [2]. Several companies listed on the capital market have suffered a heavy blow not only from the financial aspects of the company but also real and fundamental aspects such as banking and tourism companies. Even so, there are companies whose performance has improved during the pandemic, such as telecommunications companies that have had tremendous profits, because internet usage during work and studies from home is becoming higher; companies engaged in the food and beverage sector too, because these companies are producing the basic necessities that

are also currently needed. This situation causes high volatility in the capital market. The high volatility during and after the crisis was caused not only by investor sentiment but also due to uncertain economic conditions as it was not yet known when it would end [3]. The COVID-19 pandemic that occurred, in addition to causing stock market volatility, also changed investors' perspectives in making decisions and their courage to take risks [2].

The COVID-19 pandemic has had a devastating effect on world capital markets. Liu et al. [4] found that in major economies, stock markets have been affected and have dropped dramatically since the start of the COVID-19 pandemic, including countries in Asia with more negative abnormal returns than other countries. The disasters that occur have an impact on the difficulty of operational financing, increased capital costs, and tightened financial flexibility [5]. They respond to COVID-19 macroeconomic shocks through underpricing of equity risk and reducing leverage. The COVID-19 pandemic has received a significant response from the market as many activities have experienced a decline in market capitalization. The effects of COVID-19 have caused the economies of almost all countries in the world to tend to decline. This does not just have an impact on the economy; all of society is affected, which has caused dramatic changes in the way businesses act and consumer behavior [6].

Indonesia's capital market was also shaken by this virus. It can be seen from the Indonesian Composite Index (IDX), which has continued to experience a decline along with the outbreak of COVID-19 in China and spreading very rapidly to other countries in the world. The IDX composite was at the level of 5940 on 31 December 2019 and decreased to its lowest level of 4905 in June 2020 with a percentage of 17.42%. The data presented in Table 1 show the stock prices change (increase/decrease) before (1 October 2019 to 1 January 2020) and after the COVID-19 pandemic (1 February to 1 May 2020). There were 21 stocks or 87.5% of all stocks sampled in this study showing a stock prices decline and only 3 stocks or 12.5% that experienced an increase in stock prices. The decline in the IDX and most of the stock prices indicated the stock market's panic since the COVID-19 outbreak began, which had an impact on the stock market and economic growth.

**Table 1.** The decreasing of stock prices compared to the end of 2019.

| Ticker Code | Average Closing Price before COVID-19 | Average Closing Price before COVID-19 | Changing (Decreasing/Increasing) |
|---|---|---|---|
| FREN | 128.75 | 90 | −38.75 |
| ZINC | 394.5 | 197.25 | −197.25 |
| ZBRA | 63.25 | 50 | −13.25 |
| YELO | 83.25 | 50 | −33.25 |
| PWON | 570 | 394.5 | −175.5 |
| CASA | 4.625 | 4.065 | −0.56 |
| BNII | 208 | 152.25 | −55.75 |
| RMBA | 329 | 255.5 | −73.5 |
| HMSP | 2058.75 | 1665 | −393.75 |
| KLBF | 1542.5 | 1318.75 | −223.75 |
| TOWR | 760 | 833.75 | 73.75 |
| AMRT | 856.25 | 838.75 | −17.5 |
| ACES | 1621.25 | 1472.5 | −148.75 |
| TLKM | 3952.5 | 3325 | −627.5 |
| BNLI | 1203.75 | 1168.75 | −35 |
| MYOR | 2042.5 | 2001.25 | −41.25 |
| BBRI | 4290 | 3222.5 | −1067.5 |
| DNET | 3095 | 3547.5 | 452.5 |
| BBCA | 32,168.75 | 27,718.75 | −4450 |
| UNVR | 7675 | 8363.75 | 688.75 |
| BMRI | 13,456.25 | 13,000 | −456.25 |
| ASII | 6681.25 | 4511.25 | −2170 |
| SMMA | 13,887.5 | 12,937.5 | −950 |

Sources: Data Processed (2020).

This study will further find out whether COVID-19 causes a decrease in individual stock performance by using risk adjustments consisting of the Treynor, Sharpe index, and Jensen's Alpha models. The performance evaluation of portfolios is important for investors

to be more careful in choosing securities, especially during a pandemic. Investors must consider the company's performance, which may have consequences and risks [7]. Risk-adjusted performance models have been used to measure the performance of portfolios, stock price indexes, Islamic finance, and conventional indices [8,9].

The study evaluated the performance of 24 large market capitalization stocks on the Indonesia Stock Exchange which represented five price fractions due to the influence of COVID-19. The structure of the paper consists of: Section 1 as an introduction to explain the background of title selection and relevant previous research to the topic discussed. Section 2 deals with an explanation of the data used and a brief overview of the research methodology. The third section presents the results of data analysis based on the materials and methodology, which will be discussed in Section 4; Section 5 is a conclusion.

## 2. Materials and Methods

This research is a study of events that identify the effects of COVID-19 on the performance evaluation of large market capitalization shares using a risk-adjusted performance model consisting of the Treynor ratio, Sharpe Index, and Jensen's Alpha. Evaluation of portfolio or stock performance is an interesting research study object. The purpose of stock performance evaluation is to ascertain whether a stock is performing better, worse, or on a par with the benchmark. Stock performance evaluation is necessary for several reasons. First, as investors' knowledge of the shares they will own. Second, in terms of compensation, stock performance information is needed to evaluate the performance of portfolio managers. The manager's compensation is determined based on his success in managing his investment [10].

Event study is a research technique for observing the effect of an event on stock prices in the capital market. Research looks at stock returns around events that occur. An event study is also defined as a study related to the market response to an event whose information has been published to the public. Stock return movements occur around certain systematic events, especially the announcement of events that are thought to provide new information about a company as a market reaction of an event. Event study examines whether investors get abnormal returns around events and tests whether there is information content of an event or not. The market will react if an event or announcement contains information. The market will react if an event or announcement contains information. Market reactions can be observed from subsequent changes in stock prices [11]. As a major event that will determine the continuity of a company, post-merger and acquisition of large companies can cause significant long-term abnormal returns after acquisitions and mergers. This can be seen through testing several things such as initial public offering, seasoned equity offering, dividend initiation, share buyback, stock split, and reverse stock split [12].

Analysis of the response of the IDX to the presidential election events on LQ45 shares shows that the abnormal return in the event period is statistically insignificant because the market does not react, but the presidential election event affects the increase in trading volume [13]. COVID-19 is an event that has a major impact on various aspects of human life, including the economic aspects of the capital market. The economic sector is volatile—slumped. Many companies were unable to continue their operations due to the outbreak. Currently, because of the high volatility of the stock market, investors must be careful to form an optimal stock portfolio that is able to provide maximum returns and minimal risk [14]. The established portfolio and has better performance which, viewed from portfolio return and risk, can increase the likelihood of achieving investment objectives. The information contained in the announcement of these events can provide a signal in decision making for investors [15].

The investment process consists of market and performance analysis. Market analysis includes matters related to risks and expected returns, while performance analysis includes conventional and risk adjustment methods. The conventional method consists of benchmarks and style comparisons. The risk adjustment method is preferred over the

conventional method [16]. This study will focus on performance analysis based on three approaches, namely Sharpe Index, Jensen's Alpha, and Treynor, which will be explained as follows.

- Sharpe Index. The measurement uses the capital market line as a reference, namely the ratio between the standard deviation (SD) and the portfolio risk premium. SD measures the total portfolio risk or stock returns. The Sharpe index can be compared to reference stock. If the managed portfolio receives a risk premium that outperforms the market portfolio, this indicates that it is performing better than the market portfolio adjusted for total risk [17]. The Sharpe index evaluates average excess returns during volatility [14]. The average volatility of excess profits is measured by SD [5]. The Sharpe ratio method is stated in the following formula:

$$S = \frac{R_i - R_f}{\sigma_i} \tag{1}$$

Average excess return volatility is measured by the SD of excess returns [14]. The Sharpe ratio terms are stated below: where

$R_i$ = Stock return $i$;
$S$ = Sharpe ratio;
$R_f$ = Risk-free rate;
$\sigma_i$ = SD of returns of stock $i$.

- Treynor Ratio The Treynor ratio is a model used to evaluate stock performance by calculating the portfolio risk premium per unit market risk (beta). Market risk as a systematic risk, or often called beta, cannot be eliminated through diversification, that is, a trend regression of portfolio returns formed in market portfolios [5,18]. The Treynor Ratio formula is as follows [17]:

$$T = \frac{R_i - R_f}{\beta_i} \tag{2}$$

where $T$ = The Treynor Index; $\beta_i$ = The beta of the stocks.

- Jensen's Alpha Jensen's Alpha is an index of the capital market line deviation if the actual rates of return realized from the portfolio and the expected returns are not the same. This method uses the securities market line that unites the market portfolio with risk-free investment opportunities [19,20]. Jensen's Alpha also measures the difference in the average return on a portfolio or stock and the expected return. A negative (positive) index means that the portfolio provides an average return that is smaller (greater) than the expected return which consists of systematic risk, market risk premiums, and risk-free rates reflecting that investors are beating (doing worse) than the market index. The following is Jensen Alpha's formula [17]:

$$\alpha = R_i - \left[ R_f + \beta_i \left( R_m - R_f \right) \right] \tag{3}$$

where

$\alpha$ = Jensen's alpha;
$R_m$ = Market return.

This study will discuss the performance of large market capitalization stocks which are spread over several price fractions in order to represent the entire sample in the Indonesian Stock Exchange during the COVID-19 pandemic using the Treynor Ratio, Sharpe Index, and Jensen Alpha model. This model is used to see whether the research sample has a performance above average or below. To test portfolio performance evaluation, daily data from 28 February 2020 to 29 May 2020 were used which were obtained from 25 selected stocks. One sample had incomplete data so it was finally excluded from the sample; all samples in this study amounted to 24 stocks. In a pandemic period, it is expected that blue

chip stocks will have a relatively stable performance. We ran 59 IDX Composite Index data, risk-free rates, and daily stock prices. The stability and size of the company are measured based on the size of the company's capital and assets. The categorization of large cap stocks or first tier shares is in the range of ≥IDR 10 trillion. Stocks with a capitalization of between IDR 500 billion to 10 trillion are categorized as second tier stocks and for IDR 500 billion and below, they are called third tier stocks. Stocks that are in the first, second, and third tier have potential benefits, but risk-wise, blue chip stocks are relatively safer than others because the risk of value fluctuation is lower than second and third tier stocks. Second or third tier stocks tend to be cheaper, and sometimes there are times when the valuation of second tier stocks increases significantly. The price of blue chip shares per share is relatively higher than stocks in the second and third layers. For that, the capital required to invest in blue chip stocks is relatively higher. Blue chip stocks represent companies with large market capitalization. Representing the population, apart from being based on market capitalization, the sampling is also based on the price fraction or tick size [21]. The research samples are shown in Table 2.

**Table 2.** Research samples in the tick sizes.

| Tick Size (IDR) | Ticker Code | Stock Price | Market Cap |
|---|---|---|---|
| 0 s/d < 200 | FREN; LKPR; ZINC; ZBRA; YELO | 61; 131; 134; 50; 50 | 13,295.8B; 9245.05B; 3383.5B; 42,807B; 19,002B |
| 200 < 500 | PWON; CASA; BNII; RMBA; CARE | 456; 386; 234; 376; 392 | 21,960.8B; 21,027.8B; 17,834.4B; 13,686.8B; 13,034B |
| 500 to <2000 | HMSP; KLBF; TOWR; AMRT; ACES | 1445; 1495; 995; 725; 1755. | 168,080B; 70,075B; 49,849.9B; 30,105.3B; 29,988.8B |
| 2000 to <5000 | TLKM; BNLI; MYOR; BBRI; DNET | 2990; 2260; 2,410; 4,000; 3,450 | 296,196B; 63,376.5B; 53,884.5B; 490,382B; 48,938.8B |
| ≥5000 | BBCA; UNVR; BMRI; ASII; SMMA | 31,950; 7150; 6250; 5900; 15,175 | 787,727B; 295,662B; 291,446B; 238,853B; 191,097B |

Source: Indonesian Stock Exchange.

We introduce some basic concepts about performance stock, which will serve us in the following analysis of portfolio performance. Two of the most basic but important concepts are return and risk. The first step in measurement using a risk-adjusted performance model process is to calculate the individual stock and expected return that can be noted as [5,22].

$$R_i = \frac{P_t - P_{t-1}}{P_{t-1}} \tag{4}$$

where

$P_t$ = Stock price $t$;
$P_{t-1}$ = Stock price $t - 1$;
$R_i$ = Stock return $i$.

$$E(R_i) = \frac{1}{n} \sum_{t=1}^{n} R_{it} \tag{5}$$

where

$E(R_i)$ = Expected return of stock $i$;
$R_{it}$ = Future return t of stock $i$;
$n$ = The amount of data.

The next step in the portfolio formation process is to calculate the variance return ($\sigma_i{}^2$) that can be calculated using the formulation below [23]:

$$\sigma_i{}^2 = \sum_{t=1}^{n} \frac{[Ri - E(Ri)]2}{n-1} \tag{6}$$

where $\sigma_i{}^2$ = Variance return of security *i*.

Our model for market expected return $E(R_m)$ and market return ($R_{m,t}$), in this case, is the IDX composite index and can then be written as:

$$R_{m,t} = \frac{IDXC_t - IDXC_{t-1}}{IDXC_{t-1}} \tag{7}$$

where

$R_{m,t}$ = Market return *t*;
$IDXCI_t$ = Market index *t*;
$IDXCI_{t-1}$ = Market index *t* − 1.

$$E(R_m) = \frac{\sum_{t=1}^{n} Rmt}{n} \tag{8}$$

where $E(R_m)$ = Market expected return.

Then, market variance can be noted as:

$$\sigma_m^2 = \sum_{t=1}^{n} \frac{[R_m - E(R_m)]^2}{n-1} \tag{9}$$

where $\sigma_m^2$ = Variance of market return.

Beta ($\beta_i$) calculates the volatility between a security's (portfolio) return and market return. If relative market returns are related to market risk, beta shows the size of a security relative to market risk. Beta can be calculated by the following formula [24]:

$$\beta_i = \frac{\sigma_{im}}{\sigma_m^2} \tag{10}$$

where

$\beta_i$ = Beta;
$\sigma_{im}$ = Covariance among stock return and market return.

Previous research conducted prior to COVID-19 related to evaluating the performance of portfolios included in the LQ45 index on the IDX using a risk-adjusted performance model consisting of Jensen, Treynor, and the Sharpe index concluded that there was no difference in test results. However, based on the comparison of the three models used by Treynor, it shows consistent results [25]. Based on risk adjustment performance analysis on several stocks from several index institutions such as Jakarta Islamic Index (JII), LQ45, and others before the COVID-19 pandemic, it is known that there are three indices that have better performance than the risk of free market instruments and stocks [8].

## 3. Results

Table 3 presents the calculation of the expected excess return, average excess return expected, market return of variance, market deviation standard, the Jakarta composite index, and average risk-free rate return (BI rate). Based on data processing, the excess expected return and the average expected return values are −0.07463 and −0.00133, respectively, which shows that stocks returns during the COVID-19 pandemic were severely affected. Stocks underperformed and a decrease in returns occurred, meaning that the risk of investing in stocks was very high. The variance of 0.000797 shows a relatively low deviation, which means that on average, the levels of return and expected return

of securities are equal to one another. Likewise, the average Indonesian bank interest rate was at a low interest rate of around 4.5% during the COVID-19 pandemic. This is in line with the Indonesian bank's policy to reduce interest rates, which is expected to support improvements in banking intermediation and the domestic economic recovery situation that was going on during the pandemic. During the 59 trading days, the Bank of Indonesia interest rate did not change and remained at the position of 0.045 or 4.5%, meaning that during the COVID-19 pandemic, the return on the interest rate was high risk. This shows that stocks during the COVID-19 pandemic were severely affected so overall stocks underperformed; different from the period before the COVID-19 pandemic, the interest rate is at the level of 5.76% or 0.0576.

**Table 3.** Expected excess returns, average expected excess return, return of variance, deviation standard, Jakarta Composite Index, and average risk-free rate return (BI rate).

| Variable | Value | Variable | Value |
|----------|-------|----------|-------|
| $\sum_{R_m - R_f}$ | −0.07463 | $\frac{\sum(R_m - R_f)}{n}$ | −0.00133 |
| $\sigma_m^2$ | 0.000797 | $\frac{\sum(R_f)}{n}$ | 0.045491 |
| $\sigma_m$ | 0.028235 | $\sigma R_f$ | 0.027413 |
| $\sum_{(R_f)}$ | 2.5475 | $\sigma R_f^2$ | 0.000751 |

Sources: Data Processed (2020).

To find out the change in return before and after the pandemic, the difference test on the average return is carried out as presented in Table 4. The average difference test shows that the mean return difference after and before COVID-19 is insignificant, indicated by the *p*-value of 0.48 >alpha (5%) and *t* stat < *t* critical, as presented in Table 4.

**Table 4.** *t*-Test: two samples with equal variances.

| | Before COVID-19 Pandemic | After COVID-19 Pandemic |
|---|---|---|
| Mean | −0.014125221 | −0.014940522 |
| *t* Stat | 0.03357314 | |
| *p*-Value | 0.486688296 | |
| *t* Critical | 1.681952357 | |

Sources: Data Processed (2020).

Research related to the COVID-19 pandemic can be said to be an event study because the market reacts to events that occur and the market can react positively or negatively depending on what actions are taken on an event. Changes in cultural aspects (low individualism) and the government's concern for COVID-19 gave a negative signal to the movement of abnormal returns in the first week after the announcement [26]. The findings of Xu [27] prove that price movements and stock returns on the Canadian and US stock markets responded negatively and asymmetrically due to COVID-19. This information asymmetry occurs due to the negative impact of the uncertainty caused by COVID-19, although it is not significant [27].

Table 5 provides preliminary information on stock price index data during the pandemic. There are 58 days of research observations covering the entire stock price index. The descriptive showed the min value of Treynor is −8.68292 and the max is −0.286975. The Sharpe Index min value is −2.123156 and the max value is 0.114620. The Jensen Alpha ratio min value is −0.066880 and the max value is 0.043904. All methods used in this research give a negative mean. This suggests that all stocks are underperforming overall. The mean of Treynor index is −1.728652, the Sharpe method is −0.238781, and Jensen Alpha is −0.00350.

**Table 5.** Descriptive statistics of stock prices indexes.

|  | N | Mean | Std Deviation | Std Error | 95% Confident | Minimum | Maximum |
|---|---|---|---|---|---|---|---|
| Treynor | 24 | −1.728652 | 1.959693 | 0.400020 | 0.82750 | −8.68292 | −0.286975 |
| Sharpe | 24 | −0.238781 | 0.503762 | 0.102830 | 0.212720 | −2.123156 | 0.114620 |
| Jensen | 24 | −0.00350 | 0.029092 | 0.005938 | 0.012284 | −0.066880 | 0.043904 |

Sources: Data Processed (2020).

The research findings are consistent with previous empirical evidence conducted during the crisis comparing the performance of conventional and Islamic indices using risk-adjusted performance, which recommends a negative rate of return; evidence is obtained that the Islamic index performs better than the conventional [10]. The empirical evidence is also no different from other findings that have tested the market price risk response to COVID-19 using the return and equilibrium risk model. The impact of the current pandemic is a very negative response to skewness and the market price of risk. Therefore, it can be said that the current crisis is more severe than the impact of the crisis that occurred in 1987, which is based on a more negative response to the skewness slope and the total market price of risk [28]. The 2007–2008 financial crisis started to have an impact on the Romanian capital market in July 2007. An investment strategy that was considered quite successful, which was characterized by low volatility and holding period returns and positive risk adjustments, was the massive migration of bond funds to safer instruments. Performance evaluation during the crisis shows that using risk-adjusted performance consisting of the Treynor ratio, Sharpe ratio, and Jensen's alpha, all three show a positive value [29].

Table 6 presents regression results of the $\alpha$, $\beta$, $f$ statistic, and adjusted $R$ square data needed in the calculation of performance evaluation of individual stock using Treynor, Sharpe Index, and Jensen Alpha. Beta value as a slope shows the relationship between market return and individual returns which also shows a systematic risk. FREN is a stock with the highest beta value in the regression equation compared to the slope of other stocks. The beta value of FREN shows that if the market return changes by 1, it will cause the FREN return ($R_i - R_f$) change to increase by 1.8189. Adj. $R$ square of FREN shows that the contribution of the explanatory variable to the explained variable is 52%. Stocks that have no influence at all, as indicated by the highest negative adjusted R square results, are SMMA with a value of −0.0151.

FREN is one of the telecommunication companies which, in 2020, recorded an increase in the number of subscribers of 91.05% in 2019 and continued until the first quarter of 2020. Throughout the first quarter of 2020, FREN subscribers grew 46.06% compared to the same period in 2019. In line with that, in the first quarter of 2020, FREN's income also shot up 41.98% compared to the first quarter of 2019. The high slope of FREN corresponds to the extraordinary improvement in FREN's performance throughout the COVID-19 pandemic. FREN's extraordinary performance improvement throughout the COVID-19 pandemic is in line with the increasing need for online information. Therefore, the role of information technology during the pandemic has increased sharply and is much more important, in line with various restrictions imposed to suppress the spread of COVID-19. When restrictions are implemented, government and private employees from various sectors apply to work from home and ICT has been exploited maximally in the world of work, education, business, entertainment, etc. The application of digital technology in various aspects of people's lives during the COVID-19 pandemic is a solution to the limitations of direct interaction and can overcome this challenge [30].

**Table 6.** Regression result.

| Ticker Code | Coefficient | Alpha ($\alpha$) | Beta ($\beta$) | *f* Stats | Adj. *R* |
|---|---|---|---|---|---|
| FREN | c | 0.0435 | 1.8189 | 61.3990 | 0.5189 |
|  | Sign | 0.0013 | 0.0000 | 0.0000 |  |
| LKPR | c | 0.0084 | 1.1880 | 37.9760 | 0.3977 |
|  | Sign | 0.3437 | 0.0000 | 0.0000 |  |
| ZINC | c | −0.0116 | 0.9792 | 23.7289 | 0.2887 |
|  | Sign | 0.9792 | 0.0000 | 0.0001 |  |
| ZBRA | c | −0.0418 | 0.0788 | 3.5558 | 0.0436 |
|  | Sign | 0.0000 | 0.0646 | 0.0646 |  |
| YELO | c | −0.0434 | 0.0451 | 3.3507 | 0.0403 |
|  | Sign | 0.0000 | 0.0726 | 0.0726 |  |
| PWON | c | 0.0245 | 1.5810 | 64.8992 | 0.5329 |
|  | Sign | 0.0276 | 0.0000 | 0.0000 |  |
| CASA | c | −0.0388 | 0.1728 | 1.7190 | 0.0127 |
|  | Sign | 0.0000 | 0.1953 | 0.1953 |  |
| BNII | c | 0.0093 | 1.1857 | 88.8291 | 0.6106 |
|  | Sign | 0.1844 | 0.0000 | 0.0000 |  |
| RMBA | c | −0.0662 | −0.4120 | 2.2762 | 0.0223 |
|  | Sign | 0.0000 | 0.1371 | 0.1371 |  |
| HMSP | c | 0.0218 | 1.3460 | 85.0348 | 0.6001 |
|  | Sign | 0.0089 | 0.0000 | 0.0000 |  |
| KLBF | c | 0.0086 | 1.0611 | 38.4631 | 0.4008 |
|  | Sign | 0.3636 | 0.0000 | 0.0000 |  |
| TOWR | c | 0.0076 | 1.0439 | 64.5924 | 0.5317 |
|  | Sign | 0.2939 | 0.0000 | 0.0000 |  |
| AMRT | c | −0.0113 | 0.6578 | 15.6195 | 0.2070 |
|  | Sign | 0.2221 | 0.0002 | 0.0002 |  |
| ACES | c | 0.0046 | 1.0531 | 57.7231 | 0.503208 |
|  | Sign | 0.5521 | 0.0000 | 0.0000 |  |
| TLKM | c | 0.0111 | 1.2130 | 136.8926 | 0.7082 |
|  | Sign | 0.0585 | 0.0000 | 0.0000 |  |
| BNLI | c | −0.0189 | 0.5189 | 12.8854 | 0.1751 |
|  | Sign | 0.0214 | 0.0007 | 0.000706 |  |
| MYOR | c | 0.0123 | 1.1227 | 6.2602 | 0.5238 |
|  | Sign | 0.1223 | 0.0000 | 0.0000 |  |
| BBRI | c | 0.0254 | 1.5825 | 140.3481 | 0.7133 |
|  | Sign | 0.0011 | 0.0000 | 0.0000 |  |
| DNET | c | −0.0434 | 0.0420 | 0.2224 | −0.0141 |
|  | Sign | 0.0000 | 0.6391 | 0.6391 |  |
| BBCA | c | 0.0072 | 1.1589 | 142.2195 | 0.7161 |
|  | Sign | 0.1863 | 0.0000 | 0.000000 |  |
| UNVR | c | 0.0155 | 1.2273 | 76.5193 | 0.5742 |
|  | Sign | 0.0502 | 0.0000 | 0.0000 |  |
| BMRI | c | 0.0196 | 1.5124 | 120.7708 | 0.6814 |
|  | Sign | 0.0127 | 0.0000 | 0.0000 |  |
| ASII | c | 0.0192 | 1.4114 | 129.4983 | 0.6965 |
|  | Sign | 0.0070 | 0.0000 | 0.0000 |  |
| SMMA | c | −0.0461 | 0.0224 | 0.1667 | −0.0151 |
|  | Sign | 0.0000 | 0.6847 | 0.6847 |  |

Sources: Data Processed (2020).

Table 7 presents the changes in the stock price index at IDX. MYOR stock is able to provide the highest average daily return with 0.0048 and BMRI is a stock with the lowest average daily return with −0.0064. MYOR is a stock that recorded a significant growth in net profit. Based on financial report data until the third quarter of 2020, there was an increase in net profit by 42.02 percent to IDR 1.56 trillion. Profit for the year that was distributed to owners of the parent company was able to grow significantly. During the first nine months of 2020, MYOR recorded a profit of up to IDR 1.56 trillion. This amount is indeed up from the same period of the previous year, which was recorded at IDR 1.1 trillion. On the other hand, the coronavirus pandemic still puts great pressure on the performance of BMRI. As a result, the stock price of BMRI still depreciated by 24.76% during March to July 2020.

In the midst of the pressure of coronavirus or COVID-19, there are several business sectors that continue to grow and receive returns that are not different from the period before the pandemic and even increase, including businesses engaged in medical devices, pharmaceuticals, and information technology. On the other hand, there are also several sectors that have been significantly affected, even to the point of stopping their production. The influence of COVID-19 on medical stock portfolios and investor sentiment shows a positive impact on increasing portfolio returns on several exchanges in the US, Japan, China, Hong Kong, and Korea. This positive return was triggered by increased investor confidence and sentiment, both institutional investors and individual investors in the medical industry, which, during this pandemic, played a very important role [31].

**Table 7.** Risk-adjusted return performance of large market capitalization stocks for the COVID-19 pandemic period in Indonesia Stock Exchange.

| Ticker Code | $ER_i$ | $ER_f$ | $ER_m$ | $\sigma$ | $\beta$ | Treynor | Sharpe | Jensen |
|---|---|---|---|---|---|---|---|---|
| FREN | 0.0031 | 0.046 | −0.0017 | 0.053 | 1.8189 | −0.809 | −0.024 | 0.0439 |
| LKPR | −0.0022 | 0.046 | −0.0017 | 0.051 | 1.188 | −0.945 | −0.041 | 0.0085 |
| ZINC | −0.0123 | 0.046 | −0.0017 | 0.0091 | 0.9792 | −6.4 | −0.06 | −0.012 |
| ZBRA | $4 \times 10^{-5}$ | 0.046 | −0.0017 | 0.0053 | 0.0788 | −8.683 | −0.584 | −0.042 |
| YELO | $1 \times 10^{-5}$ | 0.046 | −0.0017 | 0.0613 | 0.0451 | −0.75 | −1.02 | −0.044 |
| PWON | −0.0047 | 0.046 | −0.0017 | 0.0284 | 1.581 | −1.785 | −0.032 | 0.0247 |
| CASA | −0.0014 | 0.046 | −0.0017 | 0.0429 | 0.1728 | −1.106 | −0.274 | −0.039 |
| BNII | −0.0012 | 0.046 | −0.0017 | 0.0593 | 1.1857 | −0.796 | −0.04 | 0.0094 |
| RMBA | −0.0012 | 0.046 | −0.0017 | 0.1646 | −0.412 | −0.287 | 0.1146 | −0.067 |
| HMSP | 0.0037 | 0.046 | −0.0017 | 0.0492 | 1.346 | −0.86 | −0.031 | 0.022 |
| KLBF | 0.004 | 0.046 | −0.0017 | 0.0472 | 1.0611 | −0.889 | −0.04 | 0.0087 |
| TOWR | 0.0038 | 0.046 | −0.0017 | 0.0404 | 1.0439 | −1.046 | −0.04 | 0.0076 |
| AMRT | 0.0031 | 0.046 | −0.0017 | 0.0399 | 0.6578 | −1.074 | −0.065 | −0.011 |
| ACES | 0.0003 | 0.046 | −0.0017 | 0.0419 | 1.0531 | −1.089 | −0.043 | 0.0046 |
| TLKM | −0.0007 | 0.046 | −0.0017 | 0.0409 | 1.213 | −1.143 | −0.039 | 0.0111 |
| BNLI | 0.0021 | 0.046 | −0.0017 | 0.034 | 0.5189 | −1.29 | −0.085 | −0.019 |
| MYOR | 0.0048 | 0.046 | −0.0017 | 0.0439 | 1.1227 | −0.939 | −0.037 | 0.0123 |
| BBRI | −0.0039 | 0.046 | −0.0017 | 0.0533 | 1.5825 | −0.935 | −0.032 | 0.0256 |
| DNET | 0.0002 | 0.046 | −0.0017 | 0.0189 | 0.042 | −2.424 | −1.092 | −0.044 |
| BBCA | −0.0021 | 0.046 | −0.0017 | 0.0389 | 1.1589 | −1.237 | −0.041 | 0.0072 |
| UNVR | 0.003 | 0.046 | −0.0017 | 0.0459 | 1.2273 | −0.936 | −0.035 | 0.0156 |
| BMRI | −0.0064 | 0.046 | −0.0017 | 0.0521 | 1.5124 | −1.006 | −0.035 | 0.0198 |
| ASII | −0.002 | 0.046 | −0.0017 | 0.048 | 1.4114 | -1 | −0.034 | 0.0194 |
| SMMA | −0.0016 | 0.046 | −0.0017 | 0.0117 | 0.0224 | −4.058 | −2.123 | −0.047 |

Source: Data Processed (2020).

### 3.1. Stock Performance Using the Treynor Ratio

Measuring a stock's performance using the Treynor Ratio calculation uses the average of returns and betas for a certain period. A fluctuating beta value indicates a change from the uncertain return of a stock to changes in the overall return of the market. The reason for using beta as a risk reference in investing is the uncertainty of stock prices being due to

fluctuations in market prices. If the beta value is below 1, the risk to the stock is smaller than the systematic risk. The results of the Treynor index which are positive indicate good stock performance, so the higher the index value, the better the stock performance will be. Based on the result as presented in Table 6, there is not an individual stock that showed a good performance during the COVID-19 pandemic period. This means that based on the Treynor Ratio measure, all large market capitalization stocks underperformed. The stock that has the highest negative value based on calculations using the Treynor model is ZBRA with a value of −8.683.

### 3.2. Performance Using the Sharpe Index

Measurement of stock performance using the Sharpe Index takes into account the SD of returns for each stock. If the results of the Sharpe Index calculation become bigger, then the stocks performance is becoming better. The research result as presented in Table 6 showed a negative value of Sharpe Index in almost all stocks—just one stock had a positive value, namely RMBA, with a value of 0.1146. The lowest result on the Sharpe Index calculation during the pandemic period was SMMA stock, with a value of −0.6296, while the highest was 0.1146 of RMBA stock.

### 3.3. Stock Performance Using the Jensen Alpha

Based on the Jensen Alpha method, there are 15 outperforming stocks and 9 underperforming stocks. A negative Jensen Alpha value means the stock index is under the market index. The result of the Jensen Alpha which had the smallest value during the COVID-19 pandemic periods was RMBA with a value of −0.067, and the highest was held by FREN with a value of 0.0439.

As presented in Table 8, FREN has a consistent ranking resulting from calculation using all the models, although when using Treynor, it is second best, and fourth best on the Sharpe index, indicating that this index is an effective stock and well diversified. FREN is the stock with the highest market capitalization in its tick size (third tier stocks). Recommendations that can be conveyed on the results of this study are that Jensen's Alpha is good for measuring the performance of large capitalization stocks for low price stocks. The empirical evidence is in accordance with other findings which conclude that Jensen's Alpha works well for portfolios with low price stocks [32]. There are 8 stocks from 24 stocks or about 30% which always have negative values in almost every method of measuring stock performance; these stocks are AMRT, ZINC, BNLI, CASA, ZBRA, YELO, DNET, and SMMA, indicating that these stocks underperformed during the COVID-19 pandemic period.

This empirical evidence is in accordance with the findings of research [17] which analyzed the performance of 32 stocks included in the LQ45 index during 2016–2018 using five analysis methods, namely Sortino, Information, Treynor, Sharpe, and Jensen's, concluding that not all stocks included in LQ45 have a good performance. From the 32 stocks that are included in the LQ45 index, there are 13 stocks or about 40% that always had a negative value in each method of measuring stock performance. These stocks are ADHI, AKRA, BSDE, INTP, JSMR, LPKR, LPPF, MNCN, PGAS, PTPP, SCMA, SSMS, and WIKA, and 19 stocks or about 50% had positive value.

**Table 8.** Performance ranking summary of stock price indexes.

| Ticker Code | Treynor | Ticker Code | Sharpe | Ticker Code | Jensen |
|---|---|---|---|---|---|
| RMBA | −0.287 | RMBA | 0.1146 | FREN | 0.0439 |
| YELO | −0.75 | FREN | −0.024 | BBRI | 0.0256 |
| BNII | −0.796 | HMSP | −0.031 | PWON | 0.0247 |
| FREN | −0.809 | PWON | −0.032 | HMSP | 0.022 |
| HMSP | −0.86 | BBRI | −0.032 | BMRI | 0.0198 |
| KLBF | −0.889 | ASII | −0.034 | ASII | 0.0194 |
| BBRI | −0.935 | UNVR | −0.035 | UNVR | 0.0156 |
| UNVR | −0.936 | BMRI | −0.035 | MYOR | 0.0123 |
| MYOR | −0.939 | MYOR | −0.037 | TLKM | 0.0111 |
| LKPR | −0.945 | TLKM | −0.039 | BNII | 0.0094 |
| ASII | −1 | BNII | −0.04 | KLBF | 0.0087 |
| BMRI | −1.006 | KLBF | −0.04 | LKPR | 0.0085 |
| TOWR | −1.046 | TOWR | −0.04 | TOWR | 0.0076 |
| AMRT | −1.074 | LKPR | −0.041 | BBCA | 0.0072 |
| ACES | −1.089 | BBCA | −0.041 | ACES | 0.0046 |
| CASA | −1.106 | ACES | −0.043 | AMRT | −0.011 |
| TLKM | −1.143 | ZINC | −0.06 | ZINC | −0.012 |
| BBCA | −1.237 | AMRT | −0.065 | BNLI | −0.019 |
| BNLI | −1.29 | BNLI | −0.085 | CASA | −0.039 |
| PWON | −1.785 | CASA | −0.274 | ZBRA | −0.042 |
| DNET | −2.424 | ZBRA | −0.584 | YELO | −0.044 |
| SMMA | −4.058 | YELO | −1.02 | DNET | −0.044 |
| ZINC | −6.4 | DNET | −1.092 | SMMA | −0.047 |
| ZBRA | −8.683 | SMMA | −2.123 | RMBA | −0.067 |

Sources: Data Processed (2020).

Research to identify portfolios by evaluating the performance of each LQ45 index share using three methods (Treynor, Sharpe Index, and Jensen's Alpha) did not show different results, among others. These three portfolio performance evaluation methods are the best recommendations in providing useful information, especially for fund managers as part of the investment management process [33,34]. Each evaluation method has its own characteristics: the Sharpe Index is based on total portfolio risk, the Treynor method only uses market risk, and Jensen's Alpha uses market risk. Return Sharpe ratio with market index as a benchmark is the most suitable measure, while a Sharpe ratio with the risk-free rate as the benchmark also performs not bad. Finding which method is better to choose depends on what investors' perceptions of risk are. The Sharpe method is used when investors assume that only a small portion of the portfolio returns are influenced by the market. The Treynor method is used when the investor assumes that his portfolio is well diversified. The Jensen method is used when the investor wants to find out whether the actual returns obtained and the expected returns differ when the portfolio is along the capital market lines. Besides the Sharpe ratio, the Treynor ratio performs well for a portfolio of stocks with a high price level and Jensen's Alpha performs well for a portfolio with low price stocks [25,35]. This concludes that Jensen's Alpha and Sharpe ratio are better performance measures when it comes to fund selection. The empirical findings conclude that these three models used in this research give the same result that portfolio performance is underperform. These methods cannot stand alone and are used separately because in determining an investment strategy, mutual evaluation and these methods are complementary to support each other and will obtain better results than using only one tool because of sufficient portfolio information. By using Jensen's Alpha, this study found that FREN was the stock with the highest beta with a value of 1.8189, indicating that the index was an effective and well-diversified stock. FREN is low priced and the highest market cap stock in its tick size (third tier stocks) [21]. The recommendation that can be conveyed from the results of this study is Jensen's Alpha is good for measuring the performance of large capitalization stocks for low priced stocks and this is an academic contribution that can be given.

### 4. Discussion: Performance Evaluation Portfolios and Open Innovation Dynamics

The stock market has responded negatively to COVID-19 by decreasing the number of stock trading transactions on the Indonesian Stock Exchange. The high volatility of financial markets encourages understanding and prediction of their prices in a changing market environment [36]. Facing the challenges of the global crisis due to COVID-19, it is hoped that companies will continue to strive to maintain the continuity of their business by continuing to build capabilities, including making sustainable innovations and accelerating digitalization initiatives to encourage effectiveness, business processes, optimizing the reach of market penetration, as well as introducing new innovations and services with a digital platform in the face of massive, fast, and unexpected business changes as a result of increasingly rapid advances in information and communication technology. Innovation is the successful application of new ideas born of organizational processes in different combined resources [37]. The process of creating innovation for a company is an effort that is considered burdensome and expensive. However, companies have many opportunities to make breakthroughs that can provide the best solution. Internal innovation is considered a solution that can lead to various open innovation models that take place in many companies at different levels in dealing with innovation problems. Open innovation is a dynamic approach that can create various ways through the development of ideas that involve internal companies and external parties in developing and optimally integrating new ideas for the benefit of the company in order to face highly competitive competition in the business world [38,39].

The company's openness to external factors is also more likely to have better corporate financial performance. Contribution of external factors allows companies to obtain ideas to deepen technological opportunities to improve performance [40]. Research on the need for open innovation to be applied in the financial sector shows that the open innovation model has great benefits and potential, so an open innovation approach should be applied [41]. In terms of portfolio management, open innovation enables companies to optimize resource allocation and adapt to future technological changes and opportunities for development [42]. The portfolio of open innovation dynamics deals with all things related to new products and technologies that can lead to high competitiveness [43].

The dynamics of such rapid changes have prompted the Indonesia Stock Exchange as a capital market facilitator and regulator to innovate through the application of a computerized system with modern technology that is concerned with controlling risk levels, complete trading instruments, reliable systems, and high levels of liquidity as well as improving capable trading transaction services which adjust to the very rapid changes in various aspects, especially in the face of the COVID-19 crisis situation, providing enormous benefits for investors. The dynamics of rapid change as a trigger for open innovation in improving and developing trading systems have an impact on improving stock market performance through the speed and accuracy of online data access, increasing liquidity and transparency, achieving efficiency, reducing volatility, reducing risk, and increasing investment returns and portfolio performance. As an impact of this open innovation dynamic, investors are still surviving during the COVID-19 pandemic. That innovation provides great benefits for investors, especially in times of global crisis, is also felt by Malaysian Islamic finance market investors. The innovations carried out have led to significant changes in the development of the Islamic financial market and provide transaction benefits for investors in the form of increased returns and reduced risks [44]. A dynamic approach related to evaluating portfolio performance during a pandemic can be performed by observing the reaction of the stock portfolio to fluctuations in the stock market. It is necessary to make several adjustments to obtain the optimal portfolio composition with maximum return and minimum risk. The highest return can be achieved by holding at the same risk position and adjusting its position in low- and high-risk stocks. In addition, a comparative study was also conducted to obtain the optimum portfolio composition. Characterized volatility and beta coefficients are variables that make up a portfolio profile [45].

The COVID-19 event is a cycle that keeps repeating itself and will happen again in the future; therefore, open innovation in the portfolio performance evaluation model needs to be conducted by considering dynamic risk factors. Rubera et al. [46] revealed the relationship between open innovation and new product development capabilities in portfolio development. Paulson et al. [47] use the term radical innovation for portfolio evaluation to deal with long cycle times and a high risk of uncertainty. This innovation can detect changes in the relative value of the portfolio over time which is useful for investors in evaluating performance. Flechas Chaparro et al. [48] revealed a custom innovation with four identified main features, namely treatment of uncertainty, dynamism, required input data, and management, which are interdependent. The empirical findings reveal that an approach based on different data sources and properties is more suitable for conditions with higher risk.

## 5. Conclusions

Based on data processing, the excess expected return and the average expected return values are −0.07463 and −0.00133, respectively, which shows that stocks during the COVID-19 pandemic were severely affected. This meant that almost overall, stocks underperformed and a decrease in returns occurred, meaning that the risk of investing in stocks is very high. The variance of 0.000797 shows a relatively low deviation, which means that on average, the levels of return and expected return of securities are equal to one another. FREN is a stock with the highest beta value in the regression equation compared to the slope of other stocks. The beta value of FREN shows that if the market return changes by 1, it will cause the FREN return ($R_i − R_f$) change to increase by 1.8189 based on the hypothesis test proved to be significant to explain the effect of market returns on individual returns. Adj. R square from FREN shows that the contribution of determining factors to the affected variables is 52%. Stocks that have no influence at all, as indicated by the highest negative Adjusted R square results are SMMA with a value of −0.0151. This study found that FREN found a consistent order of results, which means that the index is a good stock and is properly diversified. There are 8 stocks, or about 30%, from 24 stocks that always have negative values in almost every method of measuring stock performance. These stocks are AMRT, ZINC, BNLI, CASA, ZBRA, YELO, DNET, and SMMA, indicating that these stocks underperformed during the COVID-19 pandemic period.

Several previous studies have identified what factors can reduce stock trading volatility and increase individual stock performance through increasing stock returns during crisis periods caused by COVID-19. These factors include the participation of the government through the implementation of policies that can overcome the impact of a pandemic and reduce and suppress the spread of COVID-19 [49]. Another factor that responds to the volatility of share prices due to the COVID-19 pandemic is the cultural aspects of society, including reducing the sense of individualism and avoiding the tendency of uncertainty. Countries with a low level of individualism and high risk of rejection are able to significantly reduce the volatility of abnormal returns when compared to countries with high levels of individualism and low risk of rejection [26]. Based on the discussion section, the participation of all people, both as investors and the public in investment activities, is expected to contribute to economic recovery from the implementation of open innovation, which provides strategic steps and the best solutions during the COVID-19 pandemic. It is hoped that further analysis can be carried out on several topics as follows:

1.  The impact of the implementation of research results concerning government policies, investor and public participation, and changes in people's culture in countries in which the number of positive COVID-19 confirmations is still high on the IDX composite, which is still experiencing a decline. Then, a comparison of the level of stock volatility before and after the implementation of research can be made.
2.  Stock performance analysis using other measurement tools that have not been used in this study with an expanded sample and a comparison of the results before and after the COVID-19 pandemic.

**Author Contributions:** Conceptualisation, I.N., E.E, and R.S.A.; methodology, I.N. and R.S.A.; software, I.N.; validation, I.N., and E.E.; formal analysis, I.N. and R.S.A.; investigation, R.S.A. and L.M.; resources, I.N., E.E. and R.S.A.; data curation, I.N. and E.E.; writing—original draft preparation, I.N., E.E. and R.S.A.; writing—review and editing, I.N., E.E., R.S.A. and L.M.; visualisation, I.N. and E.E.; supervision, I.N. and E.E.; project administration, I.N.; funding acquisition, I.N., R.S.A. and L.M. All authors have read and agreed to the published version of the manuscript.

**Funding:** This research received no external funding.

**Data Availability Statement:** Data sharing is not applicable to this article.

**Conflicts of Interest:** The authors declare no conflict of interest.

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
