# Peer review of "Impact of COVID-19 on Performance Evaluation Large Market Capitalization Stocks and Open Innovation"

_2199-8531, doi:10.3390/joitmc7010056_

Round 1

Reviewer 1 Report

I read carefully the paper entitled: ”Impact of Covid-19 on Performance Evaluation of Large Market Capitalization Stocks: Evidence from Indonesia”.

  • The theme chosen by the authors has a relatively narrow horizon (…Evidence from Indonesia);
  • In my opinion, ”Risk Adjusted Ratio” (L.123-187) should not be included in (2).Literature Review.
  • If they read carefully the text of the paper, the authors will detect for themselves the negligences... But he must apply the rules of the Journal.
  • It would be good to expand the bibliography with some articles from prestigious scientific journals (WoS) published in 2019 and 2020.
  • All in all, eventually, it may be seen by an English teacher (native). 

Author Response

We send responses to reviewers' comments 

Reviewer 2 Report

Authors must make the following corrections in the paper:

-  Authors should explain better the academic contribution of the work developed. Highlighting what is innovative / original about the existing literature.

- At the end of the introduction section the authors should present the structure of the paper

- Authors should better develop the analysis to table 5 and 6.

 - Authors should develop the conclusions of the work and refer in more detail to the next steps of the work 

- Authors should add more recent references to the literature review

Author Response

(The authors gave the same response as above.)

Reviewer 3 Report

The article considers an especially relevant topic in today's society and economy. The abstract adequately expresses the objective, methodology, results and conclusion. In any case, it would be advisable to reduce it slightly so that it does not exceed the maximum of 200 words indicated by the magazine. It is recommended to provide keywords that appear in a thesaurus such as UNESCO Thesaurus (http://vocabularies.unesco.org/browser/thesaurus/en/) so that these keywords serve as a reference to locate the article. Also, keywords should not be compound phrases, unless they appear in a thesaurus because they are institutionalized. Attention must be paid to the format, as Tables 4, 6 and 7 should be duly justified and aligned with the rest of the text. The conclusions could be expanded slightly given the depth of the investigation. The list of bibliographic references should be reviewed so that the format is adapted to that required by the journal, as it appears in the rules for authors and in the template provided for the preparation of the article. It is important to carefully consult the guidelines for authors. According to the rules for authors: the initials of the authors should not be punctuated; in articles, the year must appear in bold after the journal's abbreviated name and not in parentheses after the authors ... In addition, it would be convenient to check small editorial errors as in reference 2, where there is a point in the word "Journal ".

Author Response

(The authors gave the same response as above.)

Reviewer 4 Report

The analysis has to be focused on the differences in the RAPs before and after covid considering a comparable time horizon and providing some statistical test on the differences of the covid period with respect to the previous one

Before doing the RAP analysis try to present an event study to test if the covid period is changing the risk or the return of the market and the individual stocks.

In the literature review focus on the RAP measure for the Indonesian market in order to underline the differences of your evidence with respect to the not pandemic crisis period.

Author Response

(The authors gave the same response as above.)

Round 2

Reviewer 4 Report

The new version is considering all the main issues addressed in the review report

Author Response

Thank you for your valuable comments.